# Effect of Group Housing of Preweaned Dairy Calves: Health and Fecal Commensal Antimicrobial Resistance Outcomes

**DOI:** 10.3390/antibiotics12061019

**Published:** 2023-06-06

**Authors:** Martin J. Breen, Deniece R. Williams, Essam M. Abdelfattah, Betsy M. Karle, Barbara A. Byrne, Terry W. Lehenbauer, Sharif S. Aly

**Affiliations:** 1Veterinary Medicine Teaching and Research Center, School of Veterinary Medicine, University of California Davis, Tulare, CA 93274, USA; 2Department of Animal Hygiene and Veterinary Management, Faculty of Veterinary Medicine, Benha University, Moshtohor 13736, Egypt; 3Department of Population Health and Reproduction, School of Veterinary Medicine, University of California Davis, Davis, CA 95616, USA; 4Cooperative Extension, Division of Agriculture and Natural Resources, University of California, Orland, CA 95963, USA; 5Department of Veterinary Pathology, Microbiology & Immunology, University of California Davis, Davis, CA 95616, USA

**Keywords:** group housing, bovine respiratory disease, antimicrobial resistance, broth microdilution, survival analysis

## Abstract

The objectives of this study were to investigate the effects of group housing (three calves per group) on bovine respiratory disease (BRD), diarrhea and antimicrobial resistance (AMR) to fecal commensal *Escherichia coli* (EC) and enterococci/streptococci (ES). Our study comprised two arms, one experimental and one observational. In the experimental arm, preweaned calves on a California dairy were randomized to either individual (IND; *n* = 21) or group (GRP; *n* = 21) housing, using a modified California-style wooden hutch. The study period lasted from birth to 56 days of age, during which calves were health scored daily. Cumulative incidence and hazard ratios were estimated for disease. Antimicrobial resistance outcomes were assessed using a prospective cohort design; feces were collected from each calf three times per week and EC and ES were evaluated for AMR using the broth microdilution method against a panel of 19 antimicrobial drugs (AMD). Analysis of treatment records was used to select calves that had been exposed (EXP) to an AMD-treated calf. In GRP, exposure occurred when a calf was a hutchmate with an AMD-treated calf. In IND, exposure occurred when a calf was a neighbor with an AMD-treated calf (TRT). Age-matched unexposed calves (UNEXP) were then selected for comparison. Proportions of AMR in fecal commensals among EXP, UNEXP, and TRT calves were compared between GRP and IND. Accelerated failure time survival regression models were specified to compare differences in minimum inhibitory concentration (MIC) of fecal commensals between EXP and UNEXP calves within each of GRP and IND calves separately. Group calves had a BRD hazard 1.94 times greater that of IND calves (*p* = 0.03), using BRD treatment records as the outcome. For AMR in EC isolates, higher resistance to enrofloxacin was detected in enrofloxacin-EXP GRP isolates compared with enrofloxacin-EXP IND isolates, and UNEXP GRP calves had lower resistance to ceftiofur compared with enrofloxacin-EXP and enrofloxacin-TRT calves. A significant housing-by-time interaction was detected for EC ceftiofur MIC in EXP GRP calves at 4–14 days post exposure to enrofloxacin (MIC EXP-UNEXP: µg/mL (95% CI): 10.62 (1.17, 20.07)), compared with UNEXP calves. The findings of this study show an increase in BRD hazard for group-housed calves and an increase in ceftiofur resistance in group-housed calves exposed to an enrofloxacin-treated calf.

## 1. Introduction

Antimicrobial resistance (AMR) is a major problem in both veterinary and human health. The Centers for Disease Control and Prevention (CDC) estimate that AMR causes approximately 2.8 million illnesses and 35,000 deaths per year in the US [1]. The goals of the CDC’s National Action Plan for Combating Antibiotic Resistant Bacteria are to slow their emergence and strengthen the nation’s One Health surveillance efforts [2]. One strategy to further these goals is to reduce the amount of medically important antimicrobial drugs (MIADs) used in food-producing animals. Medically important antimicrobial drugs are compounds that are utilized in human medicine to treat infections [3]. The MIADs licensed for use in cattle include cephalosporins, fluoroquinolones, macrolides and phenicols. Antimicrobial stewardship practices that limit use of MIADs to those necessary for protecting animal health and incorporating veterinary oversight and consultation into the treatment-decision-making process aim to limit the exposure of pathogenic and non-pathogenic bacteria to these compounds [4].

Commensal bacteria, such as fecal Enterobacterales and enterococci/streptococci, may harbor AMR genes that can be horizontally transmitted to potentially pathogenic bacteria [5], which may then cause infections resistant to treatment with antimicrobial drugs (AMD) in animals. In addition, there is concern that AMR commensal bacteria may infect humans via whole-bacterium transmission, whereby the AMR commensal bacteria is ingested by a human and causes disease; or the genetic elements mediating AMR may be transferred to human pathogenic bacteria via plasmids or mobile genetic elements [6,7,8]. Calf feces commonly contain AMR *E. coli* [5,6], and previous research has identified systemic AMD therapy, whether metaphylactic or therapeutic, as well as farm type, to be associated with increased fecal shedding of AMR bacteria [9,10,11].

The main causes of morbidity and mortality in preweaned dairy calves in the US are diarrhea and bovine respiratory disease (BRD) [12]; preweaned dairy calves in California (CA) have a 22% BRD prevalence [13]. Bovine respiratory disease is an important indication for AMD use in dairies, particularly for MIADs [14]. The extent to which such AMD use can select for, and alter the AMR of calves’ intestinal microbiota, is of increasing importance given the dairy industry’s stated commitment toward judicious AMD use and stewardship [15].

Over 90% of preweaned dairy calves in CA are raised in individual housing [16], either on their farm of birth (59.7%) or offsite in a dedicated calf nursery (40.3%). California’s temperate climate allows for the year-round outdoor housing of calves, most often in hutches made from wood with two dividers and slatted floors, creating a single hutch composed of three individual calf units, commonly referred to as the CA-style hutch [17]. Preventing pathogen spread is a perceived benefit of individual calf housing; however, most CA-style hutches still permit some contact between calves via the hole at the front of the hutch, allowing for the exchange of commensal and pathogenic bacteria. The effect of group housing on preweaned calf health and performance is not clear; some studies report higher incidences of respiratory disease for group-housed calves compared with individually housed calves [16,18], while others report no differences [19].

In a cross-sectional study of AMR in preweaned dairy calves housed in individual or group pens of 6–20 calves in New York [20], researchers reported increased resistance to quinolones, ciprofloxacin and nalidixic acid among group-housed calves compared to individually housed calves. Housing could affect the persistence of AMR bacteria, as group-housed calves share bacteria via calf-to-calf contact, allo-grooming, etc. In addition, a calf that receives an AMD excretes antimicrobial residues in their feces and urine, which other calves may then be exposed to. Individually housed calves also shed AMR bacteria and AMD residues, but the persistence and spread of AMR bacteria, and the transfer of AMD residues to neighboring calves, may be different compared to group-housed calves.

Given the history of CA’s ballot Propositions 2 (“Standards For Confining Farm Animals Initiative”) [21] and 12 (“The Farm Animal Confinement Initiative”) [22] that made amendments to California’s Health and Safety Code (HSC; Chapter 13.8 sections 25990–25994) concerning the space requirements for veal calves, breeding pigs and laying hens, it is possible that the group housing of dairy calves will be of future interest to consumers. Addressing consumer interests as it relates to group housing may present a substantial cost for producers, as it would require significant infrastructure and management changes. The current study aims to evaluate a solution for producers to implement group housing by removing the inner partitions of the conventional CA-style hutch, thereby creating a single group hutch. Furthermore, while AMD administration in food-producing animals is among the primary risk factors for AMR [23,24,25], the impact of such a modified group hutch on the spread of AMR bacteria in preweaned calves is unclear.

The objective of the randomized control trial arm of this study was to estimate differences in the incidence of AMD treatment, BRD and diarrhea between group- and individually housed dairy calves during the preweaned period using California wooden hutches. The objectives of the prospective cohort arm of this study were (a) to estimate the incidence of AMR in group- and individually housed dairy calves during the early preweaned period by phenotyping fecal commensal *Escherichia coli* (EC) and enterococci/streptococci (ES), and (b) to compare the acquisition of AMR in cohorts of untreated calves within each housing type using proximity to parenterally treated calves as exposure.

## 2. Results

### 2.1. Group Versus Individual Housing Trial

A total of 42 heifer calves (29 Jersey and 13 Jersey–Holstein cross) were enrolled over a period of a week in August 2021. One individually housed (IND) calf died during the study and the cause of death was not determined. Two group-housed (GRP) calves were partitioned from their hutchmates for no longer than 24 h during the study. Both calves were diarrheic and required handfeeding. Both calves received antimicrobial treatment and one was administered intravenous fluid therapy by farm staff. These calves were both sampled as TRT calves, and the hutchmates of these calves were sampled as EXP calves. Due to the short period of time that the treated calf’s hutch partition was in place relative to the period of AMR shedding that occurs following AMD treatment, no effect of partition was presumed.

#### 2.1.1. Descriptive Statistics and Disease Incidence

Baseline characteristics of the 42 study calves and GRP vs. IND comparisons showed no differences between breed, body measurements at enrollment or serum total protein measurements (Table A1). Group-housed calves had a higher cumulative incidence of BRD compared to individually housed calves based on treatment (GRP 66.7% SE 10.3; IND 47.6% SE 10.9) and scoring system (GRP 61.9% SE 10.6; IND 38.1% SE 10.6), although the differences were not significant. The cumulative incidence of diarrhea was comparable between both groups using treatment (GRP 33.3%; IND 33.3% SE 10.3) and scoring system (GRP 100%; IND 95.2% SE 4.7;). The proportion of calves treated for any reason in both groups was identical (GRP 71.4%; IND 71.4% SE 9.9). Cumulative incidences are presented in Table A2.

#### 2.1.2. Disease Hazard

The results of the Cox proportional hazards (PH) model for BRD incidence are presented in Table 1. In all health outcome models, neither breed, total protein nor week 1 body weight was identified as a confounder using the method of change in estimates. Using treatment for BRD as the censoring variable, the GRP calves had a hazard ratio of 1.94 compared to the IND calves (SE 0.589, *p* = 0.03). Using the BRD scoring system as the censoring variable, the hazard ratio was 1.84 (SE 0.329, *p* = 0.30). There were no significant differences in the diarrhea hazard among group- and individually housed calves, using both types of outcomes tested (treatment vs. fecal score).

### 2.2. Longitudinal Cohort Study for AMR Hypotheses

#### 2.2.1. Antimicrobial Drug Treatments

Three AMDs were used for treatment during the observation period: enrofloxacin (age at treatment, mean (SE): GRP; 10 days (1.58), IND; 11.3 days (0.42)), tulathromycin (age at treatment, mean (SE): GRP; *n* = 0; IND; 32.5 days (7.5)) and florfenicol (age at treatment: GRP; 20 days (-) (-), *n* = 1, IND; 18.5 days (2.12)). For enrofloxacin treatments, no other previous treatments, except for neomycin soluble powder initially added for up to 14 days to liquid feed, occurred for any of the EXP, UNEXP or TRT calves, which allowed for the cohort study on the effect of exposure to calves treated with enrofloxacin.

Of the 10 IND EXP calves, 2 were exposed to calves treated with florfenicol, and both had previously been exposed to enrofloxacin (that is, both had a neighboring calf that had been treated with enrofloxacin) with 7 and 9 days between enrofloxacin and florfenicol exposures, respectively. For tulathromycin treatments, one of two IND EXP calves had previously been exposed to enrofloxacin, with 29 days between exposures. In assembling the cohort study for the effect of exposure to calves treated with florfenicol and tulathromycin, the high incidence of AMD treatment overall (which limited our sample size, as it reduced the available pool of UENXP calves), we were unable to evaluate the effect that previous enrofloxacin exposure had on the baseline resistance of these calves for analysis of the effect of florfenicol and tulathromycin. Therefore, for these exposures (florfenicol and tulathromycin), we did not model changes in MIC and calculated frequencies of resistance stratified by treatment status only.

#### 2.2.2. Antimicrobial Resistance

EC Descriptive statistics: For EC isolates; in GRP calves, a total of 123 isolates (UNEXP: 59, EXP: 37, TRT: 27) were submitted for AMR phenotyping; fecal sample cultures from 3 EXP and 2 TRT calves did not yield any EC colonies after three attempts and were excluded. The mean age at sampling was 16.8 days (SE 0.547, min = 5, Max = 31). In IND calves, a total of 160 (UNEXP: 47, EXP: 56, TRT: 57) isolates were submitted for AMR phenotyping. Fecal sample cultures from two UNEXP, three EXP, and two TRT calves did not yield any EC colonies after three attempts and were excluded. The mean age at sampling was 21.5 days (SE 0.788, min = 8, Max = 50). Of the EC subsample that was speciated using the matrix-assisted laser desorption ionization time-of-flight mass spectrometry (MALDI-ToF MS) identification, 20/20 (100%) were identified as EC.

ES Descriptive statistics: For ES isolates; in GRP calves, a total of 117 (UNEXP: 56, EXP: 36, TRT: 25) isolates were submitted for phenotyping; 3 fecal samples from UNEXP, 4 from EXP, 4 from TRT did not yield ES and were excluded. The mean age at sampling was 16.6 days (SE 0.548, min = 5, Max = 31). For IND calves, a total of 155 (UNEXP: 42, EXP: 58, TRT: 55) isolates were submitted for phenotyping; 7 fecal samples from UNEXP, 1 from EXP and 4 from TRT did not yield ES and were excluded. The mean age at sampling was 21.4 days (SE 0.798, min = 8, max = 50). In total, 227 of the 277 (81.95%) ES isolates were speciated as Enterococcus spp., and 18.01% were Streptococcus spp., which were further attributed as 10.65% from feces of GRP calves, and 23.87% from feces of IND calves (*p* < 0.01). Within the exposure categories, the largest difference in proportions of ES confirmed as Streptococcus spp. was observed between the unexposed calves (GRP; EXP ES = Strep spp.; 13.15% UNEXP ES = Strep spp.; 3.38%; IND; EXP ES = Strep. spp.; 13.79%, UNEXP ES = Strep spp.; 40.48%).

#### 2.2.3. EC Isolates AMR Phenotype

Descriptive statistics: Baseline comparisons of the calves used for the enrofloxacin-exposed cohort are provided in Table A3. There were no significant differences between breed, total protein or body measurements at enrollment. Of the EC isolates, 91.52% were multidrug resistant (MDR). There were no significant differences in MDR between GRP and IND (GRP 91.06% (SE: 2.57%), IND 91.87% (SE: 2.16%), *p* = 0.81). For the enrofloxacin-exposed cohort, the percentages of EC isolates harvested from 1–14 days post exposure (DPE), resistant to each antimicrobial included in our analysis, are summarized in Table 2 and Table 3. There was a lower percentage of enrofloxacin-resistant isolates among GRP UNEXP compared with GRP EXP and GRP TRT. A higher percentage of GRP EXP isolates were resistant to enrofloxacin than IND EXP. Additionally, a higher percentage of ceftiofur- and florfenicol-resistant isolates were observed among IND UNEXP compared with GRP UNEXP. Proportions of resistance for calves where florfenicol and tulathromycin were the drug of treatment are presented in Table A4 and Table A5, respectively.

##### Interval Censored Parametric Accelerated Failure Time (AFT) Models of AMR in EC Isolates

For the AFT regression models, all results pertain to instances where enrofloxacin was the drug of exposure. Results for final EC AMR models that identified significant exposure by time interactions, measured using the MIC ratio, are presented in Table 4 (GRP) and Table 5 and Table 6 (IND). Effect modification for exposure by time was detected for ceftiofur resistance in isolates from among GRP calves at 4 to 14 DPE. 

A logistic regression model specified for the single MIC estimate for trimethoprim-sulfamethoxazole in GRP calves showed no significant difference between exposed and unexposed calves (OR (95% CI); 0.78 (0.16, 3.70)), but a significant effect of time was detected (OR (95% CI); 1–3 DPE: 4.67 (1.07–20.31), *p* = 0.04); 4–14 DPE: 23.47 (3.36, 164.03)). The model for gentamicin in GRP calves consisted of data where 63/66 were censored; 52/66 were right-censored and 11/66 were left-censored. Such a high frequency of censoring resulted in a poorly specified model, with point estimates exceeding the hypothetical concentration limit (10^6^ µg/mL), and it was therefore excluded. For IND calves, there was effect modification of exposure by time for ceftiofur (7–14 DPE), danofloxacin (1–5 DPE), florfenicol (1–5 DPE) and gentamicin (1–3 DPE), and exposure decreased the MIC at these time intervals.

The model-predicted MICs for ceftiofur in GRP calves are shown in Figure 1. Figures for other models are presented in Appendix A. Model parameters from significant models are presented in Table A8. 

Finally, combinations of expected mean MIC differences between EXP and UNEXP for isolates from GRP and IND models with significant interaction terms were estimated using the delta method and are presented in Table 7 and Table 8 for GRP and IND, respectively. Significant increases in mean MIC were detected for GRP EXP calves for 4–14 DPE for ceftiofur compared to GRP UNEXP calves (MIC EXP-UNEXP µg/mL (95% CI): 10.62 (1.17, 20.07)). No significant differences between IND EXP and UNEXP were detected for mean MIC comparisons. Complete model results and parameters for EC isolates and mean MIC predictions are available in the Appendix A.

#### 2.2.4. ES Isolates AMR Phenotype

Descriptive Statistics: Of all ES isolates, 33.94% were MDR, with no significant differences between GRP and IND (GRP 27.86% (SE: 4.06%), IND 38.71% (SE: 3.912%), *p* = 0.06). The proportions of resistant ES isolates, harvested 1–14 DPE with enrofloxacin, are summarized in Table 9. The proportions of ES resistance for calves where florfenicol and tulathromycin were the drug of treatment are presented in Table A6 and Table A7, respectively.

No significant differences in the proportion of ES resistance were detected between GRP and IND, or EXP, UNEXP and TRT, for any of the AMDs tested.

##### Interval Censored Parametric AFT Survival Models of AMR in ES Isolates

For the AFT regression models, all results pertain to instances where enrofloxacin was the drug of exposure. Due to the high degree of right censoring observed in the data, models for tetracycline (GRP, IND), gamithromycin (IND) and tilmicosin (GRP) either did not converge or produced estimates that exceeded the possible concentration of antimicrobial material and were therefore excluded. For GRP calves, there was a negative time effect (MIC Ratio < 1) on predicted MICs for penicillin at 6–14 DPE and ampicillin at 8–14 DPE. A positive time effect was observed for florfenicol at 4–14 DPE. There were no significant two-way interactions between exposure and DPE. In IND calves, there were no significant time or exposure effects. Complete model results and mean MIC predictions are available in the Appendix A.

## 3. Discussion

The current study findings demonstrate a higher hazard for BRD treatment among group-housed calves compared with individually housed calves. We detected a significant increase in ceftiofur resistance in EC from GRP EXP calves compared with GRP UNEXP calves. Additionally, a higher proportion of GRP EXP isolates showed enrofloxacin resistance when compared with IND EXP isolates, while a lower proportion of GRP UNEXP isolates were ceftiofur and florfenicol resistant when compared with IND UNEXP.

### 3.1. Disease Occurrence in Group- Versus Individually Housed Calves

There has been much investigation into morbidity in group-housed calves, with some studies reporting higher incidences of BRD and mortality in group-housed calves [26,27], while others found no differences [28,29,30,31,32]. The GRP calves had significantly higher hazard for BRD treatment. Given that group-housed calves share a hutch and have increased contact with their conspecifics, it is possible that one group-housed calf developing BRD increases the risk that a hutchmate will develop BRD shortly after or serve as a reservoir for infection for other calves in the hutch. We detected no significant difference in BRD hazard when using the CA BRD scoring system, which could be related to small sample size and variability introduced by multiple raters conducting the calf scoring. Three members of the research team were responsible for determining calf health outcomes for our data, and all were veterinarians trained to use the scoring systems, while one member of farm staff was responsible for administering treatments. Group housing may also have altered the temporal pattern of BRD among calves. Further, due to the nature of the experimental arm of the trial, it was impossible to blind workers or researchers to housing, which could have resulted in bias in assessing health and treatment outcomes.

A high incidence of diarrhea was observed in our study; all but one calf had a fecal score of 3 at least once. Equal numbers of calves in GRP and IND received treatment (AMD +/− intravenous fluid therapy) for diarrhea (33.33%). The incidence of treatments for diarrhea was higher than the 16% of calves treated for diarrhea in a sample of US dairy operations reported previously [14].

The cumulative incidence of BRD in the current study was higher than the 22.8% calf–caretaker treatment-as-diagnosis, or the 17.8% reported by Dubrovsky et al. (2019) using the CA BRD scoring system-as-diagnosis on five California dairy farms [33]. The current study occurred between August and September in the Central Valley of California, where summertime high temperatures regularly exceed 38 °C. During the study period, 14 days had maximum daytime temperatures above 38 °C. Differences in incidence observed in this study compared to other studies could be in part due to climatic conditions, as BRD hazard in California is increased in the summer season compared with winter [33]. There was a nonsignificant numerical difference of 23.8% in incidence of BRD in GRP- versus IND-housed calves. The lack of significance of difference in incidence of such a magnitude may be attributed to small sample size, especially given the estimated power of 23.3%. As such, further studies employing larger sample sizes are necessary.

The group housing of preweaned dairy calves differs in its management requirements; for example, treating diarrheic group-housed calves can present a challenge because of the care that they may need (oral rehydration, hand feeding, etc.). In this study, the partitions could be placed back into the hutch to help with the treatment of diseased calves. A survey of 242 smallholder dairy farmers in Brazil found that saving time and labor was a motivator for group housing calves [34]. Kung et al. (1997) reported a ten-fold reduction in management time/calf using group housing; however, that study evaluated group housing using automated feeders [35]. It is unclear if group housing represents any reduction in time or labor in intensive dairy operations, and there may be increased time and labor requirements to identify, treat and rehabilitate sick animals. Svensson et al. (2006) reported that for group-housed calves, health outcomes were worse only for calves that were kept in groups >10 calves [36]. Within the literature concerning group housing and calf health, there is an abundance of different grouping strategies, from the age at grouping to the number of calves in the group. Therefore, caution should be exercised when extrapolating results of one study to potential outcomes on a given system. Cobb et al. (2014) reported that group-housed calves had increased immunological biomarkers compared to individually housed calves, indicating better immune function [37]. In our study, we grouped calves at 7 days of age to give them time to adjust to bottle feeding [38]. Strategies that group calves later in life (3–4 weeks) coincide with high-risk periods for BRD [33], and such grouping may introduce social stressors to the calf that may inhibit its ability to fight infection.

### 3.2. Antimicrobial Resistance Phenotype of Fecal Commensals from the Study Calves

We observed high levels of multidrug resistance among EC isolates from GRP and IND, at >90%. Additionally, levels of AMR in our study were higher than those described earlier by Foutz et al. (2018) [23]. Management factors, such as feeding of waste milk and addition of neomycin to milk fed, have been associated with increased AMR in preweaned dairy calves [11]. The rates of AMD usage on the study farm were also higher (15/21, or 71.4%, of calves in both GRP and IND receiving at least one AMD treatment) when compared with treatment data from a longitudinal study of US preweaned heifer management (826/2545, or 32.5%, of preweaned heifer calves receiving at least one treatment) [12].

Complete resistance (100%) to tylosin and tiamulin were observed amongst EC isolates, using MIC resistance cutoffs adopted from Gram-negative respiratory pathogens [39]. Tiamulin is a synthetic derivative of pleuromutilin which targets the 50s subunit of the bacterial ribosome [40]. It is not labeled for use in cattle. This high level of EC resistance to tiamulin is in agreement with results from a California-wide AMR cohort study [41]. EC are intrinsically resistant to the macrolides erythromycin and azithromycin; however, we saw low levels of resistance to tulathromycin (25.18%) and tildipirosin (6.83%). Macrolides are a common antimicrobial class used to treat BRD on dairy operations [14]. EC may be a reservoir for macrolide resistance mechanisms [42] and may serve as a useful indicator of macrolide resistance in calf populations. More research is needed to better understand the mechanisms of macrolide resistance among commensal fecal EC isolated from cattle populations.

### 3.3. Association between Antimicrobial Resistance and Type of Calf Housing

We report increased resistance of EC to ceftiofur in group-housed calves at 4–14 days post exposure to an enrofloxacin-treated calf, compared with unexposed group-housed calves. This is an interesting finding given that the drug of exposure is not a cephalosporin, but a fluoroquinolone. The co-selection of antimicrobial resistance due to gene linkage is a well-described phenomenon and may be an explanation for this result [43]. Genetic determinants of fluoroquinolone resistance belonging to the *qnr* family have been associated with the presence of extended-spectrum beta-lactamase resistance, and Beyi et al. (2021) [44] reported an increase in the *oxa* beta-lactam resistance gene following the administration of high-dose enrofloxacin to 12–16-week-old calves. The high proportion of multidrug resistance detected in our study supports the hypothesis of cross-resistance as a result of genetically linked resistance determinants; however, future studies that include genetic sequencing are needed to further investigate these results. Fecal commensal AMR increasing in bovines following treatment with AMD has previously been described [9,24]; however, to the best of our knowledge, this is the first study that attempted to capture that effect in non-treated calves that occupy the same hutch. We predicted an increase in EC MIC to ceftiofur (mean MIC EXP-UNEXP (95% CI) µg/mL; 10.62 (1.17, 20.06)) in the GRP EXP calves at 4–14 days post exposure. Currently, the CLSI breakpoint for EC ceftiofur resistance is 8 µg/mL. An increase of 10.62 µg/mL may represent a clinically important shift in resistance. In an experimental study comparing AMR in EC isolated from enrofloxacin-treated piglets, De Smet et al. (2011) observed a reduction in bacterial counts and a shift in bacterial enrofloxacin AMR phenotype from predominantly wild type (MIC < 0.12 µg/mL) to majority non-wild type (>32 µg/mL) within 58 h post treatment with enrofloxacin [45]. In an observational study conducted on a commercial calf ranch, Cella et al. (2021) detected an increase in the number of ceftiofur-resistant bacteria at days 3 and 4 post treatment in ceftiofur-treated calves [9]. The 1–2 day delay in detection of increased AMR in exposed versus treated animals in the current study compared to that of Cella et al. may be due to the different methodologies used (bacterial counts on ceftiofur impregnated medium vs. broth microdilution), a lag in AMR acquisition in calves exposed to a treated calf, compared to those receiving treatment [9], or the fact that in the current study, enrofloxacin, not ceftiofur, was the AMD used to treat. A 2-day delay in uptake of fluoroquinolone resistant EC by uninoculated subjects was reported by Andraud et al. (2011) in a study assessing the potential for horizontal transmission of AMR between group-housed pigs [46]; this study had a similar objective as the current observational study but was experimental in nature. Whereas in our study, exposure was determined by proximity to an AMD-treated calf, the previous study used the inoculation of conspecific pigs with a known concentration of AMR EC as the intervention.

Interestingly, we did not observe an interaction effect of exposure and time for enrofloxacin resistance in EC isolates in our models. However, a greater proportion of GRP EXP isolates were enrofloxacin-resistant when compared with IND EXP (% resistant (95% CI); GRP EXP; 88% (75.3, 100), IND EXP; 53.6% (28.9, 78.3)). This agrees with the results of Pereira et al. (2014), who reported that preweaned individually housed calves were at decreased odds of shedding fluoroquinolone-resistant EC than group-housed calves (OR (95% CI); ciprofloxacin: 0.2 (0.1, 0.4)). One proposed reason for this finding by Pereira et al. is that group-housed calves may be more likely to receive treatment for BRD (as evidenced by the increased GRP BRD treatment hazard detected in our study) and therefore more likely to shed bacteria resistant to the AMD used to treat. However, in our study, none of the EXP calves received treatment with enrofloxacin, but they did share a hutch with a calf who had received enrofloxacin treatment.

The proportion of EC isolates resistant to ceftiofur and florfenicol in our study was higher among IND UNEXP than GRP UNEXP, and such findings are partly in agreement with those of Pereira et al., who reported that EC from preweaned individually housed calves were at increased odds of having cephalosporin-resistant EC compared with group-housed calves (OR (95% CI); 2.0 (1, 5)). It is unclear why the proportion of florfenicol and ceftiofur resistance was higher among the IND UNEXP calves in our study, as at the time of sampling, none of these calves had received any AMD treatment, except for neomycin added to the milk. These differences in resistance proportions may be reflective of dissimilarities in microbial dynamics between calves housed under different conditions and warrants further investigation. Additionally, a single EC and ES isolate were phenotyped at each timepoint, which may have introduced information bias, particularly if the fecal commensal populations of GRP and IND calves were different.

The IND AFT models showed an increase in AMR in UNEXP calves against ceftiofur, danofloxacin, gentamicin and florfenicol compared to EXP calves. This finding may reflect several underlying processes related to calf behavior, microbial dynamics and study design. For example, individually housed calves have a reduced ability to allo-groom with their neighbors, which may restrict their ability to transfer AMR genes from one hutch to another. Furthermore, the ability for IND calves to allo-groom is conditional on the calf presenting itself at the front of the hutch. The frequency and intensity of these behaviors may differ by housing types and health status; an IND calf who has been treated with an AMD (and shedding-resistant bacteria) may be less likely to have a close-contact event with their neighbor due to their illness. In contrast, IND calves that have no ill neighbors may have increased opportunity for social contact that results in the transmission of AMR bacteria or genes across the hutch barrier. More research is needed to identify management and behavioral risk factors for the transfer of AMR between calves. Additionally, the group housing cohort had 2–3 unexposed calves for every exposed calf, whereas in the individual housing cohort, the ratio was 1:1, which may have led to low power in the analysis of the outcomes from individual housing.

The observation of increased acquisition of beta-lactam AMR by untreated group-housed calves that were exposed to a treated hutchmate in the current study was based on a single California herd. Future studies are needed to ascertain the repeatability of our findings and their implications, including research on the prevention of AMR bacterial transfer in group-housed calves. Mitigation of the risk of transfer of antimicrobial resistance between treated and untreated group-housed calves may be possible via modifications in calf management, including (1) reducing the risk of administering AMD treatments against BRD by adopting BRD prevention and control measures; (2) temporary isolation of calves treated with AMD and (3) novel manure management systems to minimize exposure to fecal and urine excreta containing either resistant bacterial and/or drug residues, which should be explored for group housing systems.

### 3.4. Accelerated Failure Time Interval Regression Models to Quantify Change in Antimicrobial Resistance

The methodologies proposed to deal with the problem of interval censored MIC data include dichotomization, mixture models, survival models including proportional hazard models and accelerated time frailty models [47,48]. Each of these models has its relative merits and disadvantages. Dichotomization can be useful as a tool to determine resistance as it pertains to clinical decision making; however, clinical resistance thresholds for the specific drug, bacterial species and organ system combinations are often missing, and if they exist, they change over time, rendering prior analyses obsolete. Such dichotomization in our study dataset resulted in perfect right censoring across DPE levels of candidate logistic regression models. The use of survival models to estimate MICs has been described by Stegemen et al. (2006) [49]; however, in that research, Cox proportional hazard (PH) models, not AFT models, were used. Cox PH models assume that the effect of the covariates is multiplicative with respect to the hazard, whereas AFT models assume that the effect of the covariate is multiplicative with respect to the survival time. An advantage of AFT models is that the parametric likelihood equations for calculating model coefficients are well suited to handle left-, interval- and right-censored data. Such types of data are commonly seen in longitudinal epidemiological studies, where outcomes are diagnosed at follow-up visits. The time to event is not known precisely, but rather is known to have occurred between two time points. As the outcome of interest was the temporal and spatial relationship between calf housing and AMR, coefficients from AFT regression models could be adapted to test our hypotheses by using the interval-censored MICs in place of follow-up time. The broth microdilution method facilitated an efficient method for comparing resistance in commensal bacteria isolated from longitudinal samples of individual calves against a large panel of AMDs. A disadvantage of the standard broth microdilution technique is that bacterial MICs may exceed the concentrations of the dilution series that exist on the commercial plates used in the current study. Hence, the modeling techniques used may have led to the overestimation of MICs in some instances due to the censored nature of the MIC data. Performing end-serial dilutions would reduce the right censoring in the data and may be an important objective of future AMR research to verify predicted estimates from interval-censored models. Furthermore, Akaike Information Criteria (AIC) were used to select the best-fitting parametric distribution for the interval-censored survival models. Further research is needed to characterize the most prevalent survival distributions of AMR in preweaned dairy calves following AMD exposure.

## 4. Materials and Methods

### 4.1. Group Versus Individual Housing Trial

The study protocol was approved by the University of California Davis Institutional Animal Care and Use Committee (Protocol #21748). A total of 42 calves on a large commercial dairy located in the Southern San Joaquin Valley of California were enrolled as part of a study on the effects of group housing on behavior, activity and welfare. Methods and outcomes for the growth, activity and welfare outcomes of the study are subjects of a future report. Enrollment took place over one week in August 2021, and the study period lasted 56 days from enrollment. Calves were enrolled if they were females, of the Jersey or Jersey x Holstein breed, less than 24 h old, and had not been moved from the maternity area to be housed in a hutch. Calves were evaluated for attitude and general health before enrollment by a veterinarian (MB) and were excluded if they had obvious morbidities or congenital defects. Calves were randomized to either GRP or IND in blocks of 6 to ensure even distribution of age between the two groups. Randomization was achieved using a random number generator in Excel (Microsoft, Redmond, WA, USA). Calves were fed 2 L of heat-treated colostrum within 6 h of birth, at 12 and 24 h of life per farm protocol. Calf navels were dipped in a 7% iodine solution at birth and again at the second colostrum feeding. In addition, calves received a dose of intranasal modified live vaccine containing PI-3, BRSV and IBR (Inforce3, Zoetis, Kalamazoo, MI, USA) at 3 days of age.

Body measurements were taken so that baseline comparisons between groups could be made. Study calves were weighed using a digital weight scale (MTI Weigh Systems Inc, North Kingstown, RI, USA) at 1 week post enrollment. Withers and hip height were recorded using a measuring stick (Nasco, Fort Atkinson, WI, USA) with the calf standing on a flat even surface on the day of enrollment. Chest girth was measured on the day of enrollment using a weight tape placed around the calf’s chest just behind the elbows (Nasco, Fort Atkinson, WI, USA). All measurements were performed by one member of the research team to reduce variability (MB). Calves were housed individually in 3-compartment wooden hutches (2.5 m × 1.5 m) that were placed on concrete blocks over a concrete flush channel. Before study commencement, hutches were washed with water and a brush, sprayed with lime solution and allowed to dry in the sun.

Blood was collected for serum total protein measurement via jugular venipuncture using the Vacutainer system (BD, Franklin Lakes, NJ, USA) 48 h after the calves’ last colostrum feeding. Blood samples were transported to the Dairy Epi Lab (VMTRC, Tulare, CA 93274, USA), where they were centrifuged at 3000 RPM for 5 min. A single drop of serum was placed on a refractometer and the total protein concentration was read from the scale.

From days 1 to 3 and 16 to 21, calves were bottle fed 1.9 L TID transition milk that was supplemented with milk replacer (MB Nutritional Sciences, Lubbock, TX, USA) to a brix of 12.5%, 45 g of bovine IgG (MB Nutritional Sciences, Lubbock, TX, USA), with 22 mg/kg/day of neomycin sulfate (Neomed 325, Bimeda, Oakbrook Terrace, IL, USA) added daily up to day 14. From days 3 to 15, calves were fed 1.9 L transition milk BID (supplemented as above) and received 1.9 L oral electrolyte solution (MB Nutritional Sciences, Lubbock, TX, USA). From days 22 to 45, calves were fed 1.9 L hospital milk TID with 6 g of yeast supplement (MB Nutritional Sciences, Lubbock, TX, USA). From days 45 to 60, calves were fed 1.9 L pasteurized hospital milk BID. Calves were offered calf starter and water ad lib throughout the study period and weaned at 60 days.

### 4.2. Sample Size Calculation

The sample size calculation was performed using inputs for incidence of BRD in preweaned calves in CA from Dubrovsky et al. [33]. The expected proportion of individually housed calves diagnosed with BRD was estimated to be 22.5%, and group housing was estimated to have a BRD risk 2.75 times that of individual housing [50]. The confidence was set at 95% and power at 80%. Using the formula provided by Wang et al. (2014) [51] (Equation (1)), the sample size necessary for each group was 21 calves.
(1)$$n={\left(Z_{\frac{\alpha }{2}}+Z_{\beta }\right)}^{2}\cdot \left(p_{1}\left(1-p_{1}\right)+p_{2}\left(1-p_{2}\right)\right)/{\left(p_{1}-p_{2}\right)}^{2}$$

Equation (1) is the sample size calculation equation for non-equality tests of proportions in two groups, where *Z*_*a*/2_ is the critical value of the Normal distribution at *α*/2, *Z_β_* is the critical value of the Normal distribution at B and *p*_1_ and *p*_2_ are the expected sample proportions of the two groups.

### 4.3. Hutch Modifications

The standard 3-compartment wooden hutch was modified for the GRP hutches before the study period. The partitions were altered so that they could be lifted out as individual boards, therefore creating one large common hutch for 3 calves. On day 7–8 post enrollment, partitions in the GRP hutches were removed and stored near the hutches. If a calf needed to be isolated to receive treatment or required hand feeding (determined by farm staff), one partition was placed back into the hutch to create one individual pen within the group hutch. If the calf resumed feeding within 24 h, the partition was removed. Otherwise, the calf was evaluated for vigor and appetite at every subsequent feeding by researchers until it was deemed healthy enough to join its hutchmates. Although individual pens had partitions, calves still had opportunities for physical contact with neighbors through the front openings of the hutch.

### 4.4. Treatment Data

Treatment data were extracted from on-farm computer software (DairyComp 305, Valley Agricultural Software Inc., Tulare, CA, USA) to identify AMD treatments on the study calves. Study personnel had no input on disease diagnosis or treatment decisions for the purpose of medical treatment; instead, all treatments were carried out by farm staff as per the farm health protocol developed by the herd veterinarian. Treatments were recorded by staff at time of treatment using a handheld pocket computer and RFID scanner.

### 4.5. Data Collection

In addition to on-farm treatment records, calves were assessed daily for health by study personnel. Bovine respiratory disease was evaluated using the California BRD scoring system [52]. Diarrhea was assessed using a 3-point fecal consistency scale (1 = Normal, 2 = Loose, 3 = Watery) [53] by visual observation of recent feces in the hutch, and a fecal score of 3 was considered diarrhea.

Fecal samples were collected from all study calves three times per week (Monday, Wednesday, Friday). Sterile lube was applied to a gloved finger and gently inserted into the anal sphincter to stimulate defecation. Feces were collected into 50 mL polypropylene snap-top containers. Fecal consistency was scored at each sampling as described above. Gloves were changed between every calf to avoid cross-contamination. Fecal sampling was always carried out by two members of the research team, so that the individual collecting feces from the GRP calves did not enter the IND hutches. Feces were placed in a cooler with ice packs and transported to the Dairy Epi Lab for same-day processing and storage.

### 4.6. Isolation and Storage of Escherichia coli (EC)

Fresh fecal samples were plated directly onto EC Chromoselect agar with MUG (Sigma-Aldrich, St. Louis, MO, USA) for EC isolation and incubated at 44 °C for 18 h. Identification of EC was conducted by their characteristic color on the selective agar medium. Selected colonies (*n* = 2) were picked at random and subcultured onto tryptone soy agar with 5% sheep blood (SBA) (Remel, Lenexa, KS), incubated at 37 °C for 24 h, and pure colony isolates were then stored in 70% glycerol w/TSB at −80 °C until further testing after the end of the trial.

### 4.7. Isolation and Storage of Enterococci and Streptococci (ES)

Fresh fecal samples were plated directly onto the Rapid Enterococci Chromoselect agar (Sigma-Aldrich, St. Louis, MO, USA) and incubated at 35 °C for 24 h. A total of 4 pin-point blue colonies were selected from each sample, subcultured and stored as described for EC isolates. A total of 1 g of feces from each sample was also placed in 70% TSB glycerol and stored at −80 °C.

### 4.8. Bacterial Identification

Previous work has demonstrated that >90% of isolates from EC Chromoselect agar are *E. coli* without further identification [41]; therefore, a random subset (*n* = 20) of the entire presumptive EC repository was submitted for MALDI-ToF MS (Microflex^®^ LRF, Bruker Scientific LLC, Billerica, MA, USA) identification (CAHFS, Tulare, CA 93274, USA) confirmation. However, only approximately 45% of isolates from Enterococci Chromoselect agar are *Enterococcus* spp. [41]. For presumptive ES isolates, isolates in storage were sent to the MALDI-ToF MS in sequence until an *Enterococcus* spp. colony was identified. If no *Enterococcus* spp. colonies were identified, fecal samples from the freezer were replated onto Rapid Enterococci Chromoselect agar, and 2 new presumptive colonies were tested again for a total of 6 colonies per fecal sample using the same procedure. If none of the 6 colonies were *Enterococcus* spp., but instead *Streptococcus* spp., they were selected for antimicrobial resistance phenotyping.

### 4.9. Longitudinal Cohort Study for AMR Hypotheses

In order to achieve our AMR research objectives, it was necessary to identify calves in GRP and IND that were hutchmates (GRP) or neighbors (IND) with a calf treated parenterally with an AMD. Treated calves (TRT) received AMD for calfhood diseases such as BRD and diarrhea, and all AMD treatments were administered by farm staff in accordance with their protocols, developed by the herd veterinarian. Using dairy treatment records, we identified hutchmates or neighbors of TRT as exposed (EXP). For IND calves, unexposed calves could have been adjacent to the exposed calf, but not the treated calf. For each EXP calf, we then selected an unexposed age-matched calf (UNEXP). Isolates from EXP and TRT calves, obtained from 5 days before exposure to 14 days after exposure or until they themselves became treated, or a second calf in the hutch became treated (GRP) were identified for AMR phenotype testing. Age matching was specified for UNEXP calves because study calves were enrolled over a period of 7 days, and the AMR microbial population of the neonatal calf microbiota is highly variable early on in life [9]. Isolates for UNEXP calves were obtained during the age period corresponding to their matched EXP set of calves. For GRP calves, UNEXP calves came from different hutches. Given these criteria, we identified 5 TRT calves from 4 hutches in GRP (1 hutch had 2 calves treated on the same day) and 7 EXP hutchmates (GRP EXP; *n*; 2 + 2 + 2 + 1 = 7). Four GRP hutches with no treated calves provided 12 UNEXP calves (GRP UNEXP; *n*; 3 + 3 + 3 + 3 = 12; one group hutch served as UNEXP twice, once where enrofloxacin was the drug of exposure for the EXP calves, and once where florfenicol was used.). For IND, 10 TRT calves were identified. Their immediate neighbor, if previously untreated in the last 7 days, was declared EXP (IND EXP; *n* = 10). An untreated IND calf that did not have a treated neighbor was selected as UNEXP (IND UNEXP; *n* = 10). For each EXP calf, the AMD that declared exposure was recorded. For a visual representation of the enrollment criteria, see Figure A1 and Figure A2.

#### Antimicrobial Resistance Phenotyping

Antimicrobial resistance phenotype was assessed using the broth microdilution method [39]. Stored EC and ES isolates were plated onto SBA and incubated at 37 °C for 24 h. Using an inoculating loop, a small amount of the growth was added to 5 mL of demineralized water and vortexed until 0.5 McFarland optical density was reached, measured by a nephelometer (ThermoFisher, Waltham, MA, USA). For EC isolates, 10 µL was added to 11 mL Mueller–Hinton (MH) broth and inverted 5–7 times. For ES isolates, 30µL was added to 11 mL MH broth. For EC and ES isolates, 50 µL of MH broth was then added to each well of a BOPO7F plate (ThermoFisher, Waltham, MA, USA) using an automatic inoculating machine (ThermoFisher, Waltham, MA, USA). In addition, 1 µL from the control well was streaked onto SBA and incubated at 37 °C for 24 h as a control. Plates were sealed with a transparent adhesive cover and incubated for 24 h at 37 °C, and AMR phenotype was evaluated using the VisionReader (ThermoFisher, Waltham, MA, USA). If the control SBA plate yielded no growth or was contaminated, or the multi-well plate showed no growth or was contaminated, the phenotyping was repeated. If this happened a second time, the isolate was excluded. The BOPO7F plate comprised 19 antibiotics: ampicillin, ceftiofur, clindamycin, danofloxacin, enrofloxacin, florfenicol, gamithromycin, gentamicin, neomycin, penicillin, sulphadimethoxine, spectinomycin, trimethoprim-sulfamethoxazole, tetracycline, tiamulin, tilmicosin, tildipirosin, tulathromycin and tylosin tartrate. Minimum inhibitory concentration (MIC) was recorded as the lowest concentration of the AMD that inhibited bacterial growth.

Isolates were classified as either resistant or susceptible based on MIC breakpoint values from CLSI (Table A9) [39]. Intermediate isolates were classified as susceptible. If no CLSI breakpoint was available, MIC breakpoints used by Abdelfattah et al. (2021) were adopted [41]. Enterobacterales are intrinsically resistant to clindamycin, penicillin and macrolides, while *Enterococcus* spp. are intrinsically resistant to cephalosporins, lincosamides, aminoglycosides, fluoroquinolones and folate pathway antagonists [39]. Drugs from these classes were excluded from the analysis of MICs for the respective organisms. EC were classified as MDR when resistance was detected to at least one drug in three antimicrobial classes: aminopenicillins, cephalosporins, fluoroquinolones, amphenicols, tetracyclines, aminoglycosides and folate pathway antagonists. ES were classified as MDR when resistant to at least one drug in three or more of the following antimicrobial classes: penicillins, amphenicols, tetracyclines, pleuromutilins and macrolides. EC ATCC 25922, EC ATCC 35218 and *Enterococcus faecalis* ATCC 29,212 (American Type Culture Collection, Manassas, VA, USA) were phenotyped at least once a week during phenotyping using the broth microdilution method for quality control purposes. All quality control isolates were within the appropriate MIC ranges throughout the study.

### 4.10. Statistical Analyses

#### 4.10.1. Baseline Comparisons and Health Outcomes

Initial measurements of calf weight, hip height, withers height and chest girth were assessed for normality by visually checking quantile–quantile plots and histograms. In order to ensure the effectiveness of the randomization of calves into GRP and IND, baseline comparisons of breed distribution and body measurements were made with the *t*-test for normal variables, and the Wilcoxon rank sum test for non-normal variables. Cumulative incidence of treatment for BRD and diarrhea was compared between housing groups using Z-test for proportions in Stata 17 software.

#### 4.10.2. Survival Analysis Models for Disease Hazard

Comparisons of BRD hazard between group- and individually housed calves were performed using Cox PH models after assessing the proportionality of the baseline hazards assumption, with separate models for BRD diagnoses using the CA BRD scoring system and farm BRD treatment records. Time at risk commenced at the grouping (7–8 days) for group-housed calves, and at 7 days for individually housed calves, to capture the effect of group vs. individual housing. The proportionality of hazards assumption, that the hazard ratio of being diagnosed with BRD between GRP and IND was independent of time, was assessed using the Schoenfeld test, where the scaled Schoenfeld residuals for covariates of non-censored animals were regressed over time. A significant non-zero slope identifies variables that violate the PH assumption [54]. Robust standard error estimates were calculated with a clustered sandwich variance estimator that allowed for intra-hutch correlations. A manual model-building process was undertaken by offering the variables breed and total protein to the model containing housing type. Potential confounders were assessed by the method of change in estimates, where a change in estimates of 20% on the dependent variable of interest (housing type) upon inclusion of the potential confounder in the model provided evidence for its status as a confounder [55]. All possible two-way interactions between variables were evaluated for significance at the 5% level. Model fit was assessed by comparing AIC, where AIC = −2 natural logarithm (likelihood) + 2 × model degrees of freedom. Models with a lower AIC had a better fit.

#### 4.10.3. EC and ES AMR Phenotype

Minimum inhibitory concentrations for all EC isolates were exported from the Sensititre software and inserted into a relational database (Microsoft Access, Microsoft, Redwood, WA, USA). Percentages of resistant isolates collected 1–14 days post exposure were calculated for TRT, EXP and UNEXP calves in GRP and IND. Proportions of ES isolates that were speciated as *Streptococcus* spp. across housing types and exposure status were calculated. Resistance proportions for EC and ES isolates were compared across housing types and exposure status using the Z-test after adjusting their standard errors for repeated measures using the intra-class correlation (ICC) coefficient. The ICC was estimated by fitting a variance components model, a random effects-only model with a logit link and calf as a random intercept, separately for each AMD outcome. The variance estimate of the random effect (${\hat{\sigma }}^{2})$ was then used to compute the ICC using the formula from Wu et al. (2012) [56]:(2)$$ICC=\frac{{\hat{\sigma }}^{2}}{{\hat{\sigma }}^{2}+(\pi^{2}/3)}$$

Equation (2): Intra-class correlation coefficient for dichotomous outcome variables.

#### 4.10.4. Survival Interval Regression Models for Effect of Exposure to a Treated Calf

For modelling of the differences in MICs between EXP and UNEXP, age-matched sets of calves were analyzed separately according to the AMD that declared exposure. For antibiotics with only 1 MIC dilution on the BOPO7F plate (sulphadimethoxine, trimethoprim/sulfamethoxazole), logistic regression models were built for the outcome (presence or absence of AMR to the respective drug), to examine the association between exposure status, days relative to treatment and their two-way interaction. Random-effect models were attempted to account for repeated measures on an individual calf; however, if models did not converge or confidence intervals were inestimable due to small sample size, non-random-effect models were specified after adjusting for repeated measures using robust sandwich estimation of the standard errors [57]. For the random-effect models, the calf was specified as the random effect. For all models, exposure status and days post exposure (DPE) were offered as fixed effects.

For antibiotics with more than 1 dilution series on the BOPO7F plate, interval censored models were considered. Interval regression techniques can be used when the exact dependent variable estimate is not known but rather is known to fall within a range of two values. Using the broth microdilution method for AMR phenotyping, the MIC is the lowest concentration at which there was no bacterial growth. Hence, the actual concentration of antimicrobial material at which the organism would be inhibited falls between the recorded MIC and the previous dilution. As a result, the study isolates’ MIC data were transformed from y_i_ to [y_m_, y_i_], where y_i_ is the MIC of isolate y, and m is the previous dilution. If no drug dilutions showed growth, the dependent variable became [0, y_min_]. If all drug concentrations showed growth, the dependent variable became [y_max_, ∞]. However, MIC data are commonly right censored, raising concerns regarding the use of regular interval regression models which assume interval-censored data from a normal distribution.

For each AMD to be tested, AFT parametric survival models with interval censoring were specified. Parametric AFT models allow for the linearization of the model equation by specifying a distribution that is known to be related to the survival time under investigation, or in this case, the distribution of MIC. In the standard use case, i.e., where survival time is the dependent variable, an exponentiated coefficient (i.e., time ratio or the acceleration factor) >1 implies that exposure increases survival time; however, in the application of the AFT models to MIC data such as in this study, the exponentiated coefficients do not represent a time ratio, but rather an MIC ratio. That is, an MIC ratio >1 indicates a positive effect on the MIC for that variable, and similarly, an MIC ratio ˂1 indicates a negative effect on the MIC. A beneficial quality of AFT models is that the outcome, the acceleration factor, can be easily transformed into survival time [54], which in our case allowed us to interpret model predictions as MIC estimates on a continuous scale rather than ranges within the original dilution series reported.

A manual model-building process, as described for earlier models, was specified separately for each of the EC and ES datasets to examine the effect of exposure (EXP or UNEXP) on AMR within each housing system. Exposure status (EXP or UNEXP), sampling day relative to exposure (DPE) and their two-way interactions were tested in each model. The variable DPE was specified as a continuous variable and as a categorical variable with three or four separate time-category levels. All categorical DPE variables included DPE < 1 as its own category (pre-treatment/exposure) and 2-day intervals, 3-day intervals and 5-day intervals were offered separately to the model. Categories that resulted in a balanced number of isolates in each category, and those that limited the proportion of right censoring in each category, were prioritized, and their respective models were fitted using exponential, Weibull, generalized gamma, log–logistic and lognormal distributions. The best-fitting distribution–variable combination was identified by fitting each separately and selecting the fixed effects model that had the lowest AIC and did not have estimates that exceeded the hypothetical maximum concentration available for the MIC (>1,000,000 µg/mL). Two-way interactions between exposure status and DPE were assessed using statistical significance testing. Robust standard errors using clustered sandwich estimators were estimated, with clustering occurring at the level of the individual calf [57]. Mean MIC predictions were obtained from the sum of the exponentiated combinations of respective model coefficients using the delta method to estimate their variance and 95% confidence intervals.

Statistical significance was assessed at the 5% level of significance. All statistical analyses were carried out using Stata 17 (Stata Corp, College Station, TX, USA).

## 5. Conclusions

The current study demonstrated an increased hazard of BRD treatment in group- housed calves compared with individually housed calves. In addition, an increase in shedding of AMR commensal bacteria against ceftiofur in the feces of untreated group-housed calves housed together with a treated calf was observed. If group housing is to be adopted as a preweaned-calf-raising strategy in intensive US dairy production systems, it may be important to understand how these increased risks of disease and AMR can be mitigated. More research is needed to assess these hypotheses under different management and climate conditions to lend them broader generalizability to the dairy industry. Furthermore, the levels of AMR in EC and ES isolates observed in this study underscore the importance of antimicrobial stewardship outreach and education in calf raising.

## Figures and Tables

**Figure 1 antibiotics-12-01019-f001:**
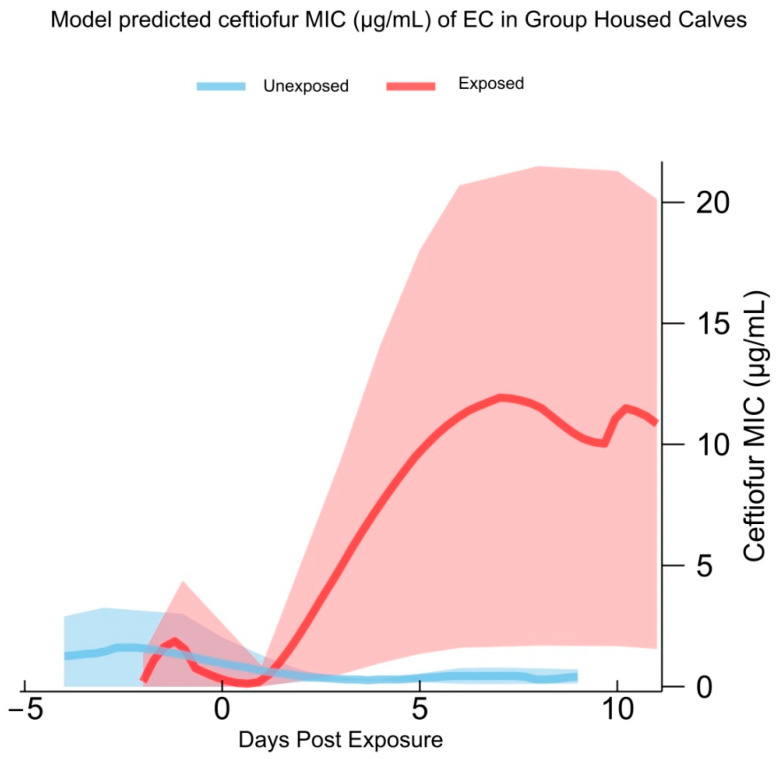
Model-predicted ceftiofur minimum inhibitory concentration (MIC) of *E. coli* (EC) isolates harvested from the feces of group-housed calves, collected from 5 days before to 14 days post exposure to an enrofloxacin-treated hutchmate. Isolates from unexposed calves were collected in the same age period from calves with no exposure to an enrofloxacin-treated calf. Shaded areas represent 95% confidence interval of model predictions.

**Table 1 antibiotics-12-01019-t001:** Disease hazards by housing type: Results from final Cox proportional hazards models for the outcomes of bovine respiratory disease (BRD) and diarrhea as identified by treatment records or diagnosed by scoring systems. The CA BRD scoring system, and a three-point fecal consistency scoring system (diarrhea = fecal score 3), were used for BRD and diarrhea, respectively. Hazard ratios, standard error (SE), *p*-values and 95% confidence intervals are shown.

Disease	Outcome	Variable	Level	Hazard Ratio	SE	*p*	95% Confidence Interval	
							Lower	Upper
BRD	Treatment	Housing	Individual	Referent				
			Group	1.94	0.589	0.03	1.07	3.52
	California BRD Scoring System ≥ 5	Housing	Individual	Referent				
			Group	1.85	0.88	0.2	0.723	4.721
Diarrhea	Treatment	Housing	Individual	Referent				
			Group	0.71	0.298	0.42	0.31	1.62
	Fecal Score = 3	Housing	Individual	Referent				
			Group	1.25	0.345	0.42	0.73	2.15

**Table 2 antibiotics-12-01019-t002:** Percent of *E. coli* (EC) isolates resistant to antimicrobials for cohort of calves by antimicrobial drug (AMD) exposure status, where exposure was defined as exposure to a hutchmate (in group-housed calves) or neighbor (in individually housed calves) treated with enrofloxacin as the AMD. Percent resistant (95% confidence intervals) for group-housed and individually housed calves cultured from fecal samples collected 1–14 days post exposure are shown. *p* values represent statistical comparisons across the row. Results in columns that have the same superscript are significantly different from each other (*p* < 0.05).

Antimicrobial Drug Tested	MIC (µg/mL) Breakpoint	Status	Group		Individual		*p*
			N	% Resistant (95% CI)	N	% Resistant (95% CI)	
Ampicillin	≥16	Unexposed	29	79.3 (52.7, 100)	20	90 (76.9, 100)	0.47
		Exposed	25	96 (88.3, 100)	28	89.3 (77.8, 100)	0.36
		Treated	20	100 (-)	27	92.6 (82.7, 100)	0.21
Ceftiofur	≥8	Unexposed	29	17.2 (3.5, 30.9) ^a,b^	20	60 (38.5, 81.5)	<0.01
		Exposed	25	60 (40.8, 79.2) ^a^	28	57.1 (23.6, 90.7)	0.89
		Treated	20	55 (33.2, 76.8) ^b^	27	55.6 (36.8, 74.3)	0.97
Enrofloxacin	≥2	Unexposed	29	79.3 (54.1, 100)	20	80 (47.5, 100)	0.97
		Exposed	25	88 (75.3, 100)	28	53.6 (28.9, 78.3)	0.02
		Treated	20	100 (-)	27	88.9 (77.0, 100)	0.12
Florfenicol	≥8	Unexposed	29	37.9 (17.5, 58.4)	20	80 (60.9, 99.1)	<0.01
		Exposed	25	40 (17.1, 62.9)	28	71.4 (48.1, 94.8)	0.07
		Treated	20	60 (35.3, 84.6)	27	66.7 (44.1, 89.3)	0.75
Tetracycline	≥8	Unexposed	29	86.2 (65.9, 100)	20	90 (76.9, 100)	0.76
		Exposed	25	96 (88.3, 100)	28	89.3 (77.8, 100)	0.36
		Treated	20	100 (-)	27	92.6 (82.7, 100)	0.21
Danofloxacin	≥1	Unexposed	29	86.2 (73.7, 98.8)	20	80 (47.5, 100)	0.71
		Exposed	25	88 (75.3, 100)	28	60.7 (42.6, 78.8)	0.03
		Treated	20	100 (-)	27	92.6 (82.7, 100)	0.21

^a,b^ Values with the same superscript are significantly different from each other. (-) 95% Confidence Interval could not be calculated.

**Table 3 antibiotics-12-01019-t003:** Percent of *E.coli* (EC) isolates resistant to antimicrobials for a cohort of calves by antimicrobial drug (AMD) exposure status, where exposure was defined as exposure to a hutchmate (in group-housed calves) or neighbor (in individually housed calves) treated with enrofloxacin as the AMD. Percent resistant isolates (95% confidence intervals) for group-housed and individually housed calves cultured from fecal samples collected 1–14 days post exposure are shown. *p* values represent statistical comparisons across the row. Results in columns that have the same superscript are significantly different from each other (*p* < 0.05).

Antimicrobial Drug Tested	MIC (µg/mL) Breakpoint	Status	Group		Individual		*p*
			N	% Resistant (95% CI)	N	% Resistant (95% CI)	
Gentamicin	≥16	Unexposed	29	75.9 (48.5, 100)	20	70 (27.9, 100)	0.82
		Exposed	25	96 (88.3, 100)	28	85.7 (70.1, 100)	0.25
		Treated	20	100 (-)	27	92.6 (82.7, 100)	0.21
Neomycin	≥8	Unexposed	29	89.7 (78.6, 100)	20	90 (76.9, 100)	0.97
		Exposed	25	96 (88.3, 100)	28	89.3 (77.8, 100)	0.36
		Treated	20	100 (-)	27	96.3 (89.2, 100)	0.38
Spectinomycin	≥64	Unexposed	29	41.4 (23.5, 59.3)	20	65 (33.6, 96.4)	0.22
		Exposed	25	44 (3.6, 84.4)	28	67.9 (50.6, 85.2)	0.29
		Treated	20	60 (22.9, 97.1)	27	59.3 (33.3, 85.3)	0.97
Trimethoprim– sulfamethoxazole	>256	Unexposed	29	79.3 (52.7, 100)	20	90 (76.9, 100)	0.47
		Exposed	25	88 (75.3, 100)	28	78.6 (63.4, 93.8)	0.36
		Treated	20	95 (85.5, 100)	27	96.3 (89.2, 100)	0.83
Sulphadimethoxine	≥64	Unexposed	29	89.7 (78.6, 100)	20	95 (85.5, 100)	0.50
		Exposed	25	100 (-)	28	92.9 (83.3, 100)	0.17
		Treated	20	100 (-)	27	96.3 (89.2, 100)	0.38

(-) 95% Confidence Interval could not be calculated.

**Table 4 antibiotics-12-01019-t004:** Results from significant models for *E. coli* (EC) from group-housed calves where enrofloxacin was the antimicrobial drug of exposure. Coefficients represent minimum inhibitory concentration (MIC) ratios for ceftiofur for fixed effects of exposure, days post exposure (DPE) and their two-way interactions. MIC ratios from parametric accelerated failure time models are estimated for each variable as the ratio of the exponentiated coefficient of one level of the variable to its referent.

Variable	Ceftiofur	
	MIC Ratio (95% CI)	*p*
Intercept	1.29 (0.36, 4.68)	0.69
Unexposed	Referent	
Exposed	0.51 (0.04, 6.69)	0.61
Pre-Exposure	Referent	
1–3 DPE	0.28 (0.08, 1.04)	0.06
4–14 DPE	0.33 (0.07, 1.49)	0.15
Exposed × Pre-Exposure	Referent	
Exposed × 1–3 DPE	3.52 (0.21, 59.81)	0.38
Exposed × 4–14 DPE	50.86 (3.55, 729.21)	<0.01

**Table 5 antibiotics-12-01019-t005:** Results from significant models for *E. coli* (EC) from individually housed calves where enrofloxacin was the antimicrobial drug of exposure. Coefficients represent minimum inhibitory concentration (MIC) ratios for ceftiofur and danofloxacin for fixed effects of exposure, days post exposure (DPE) and their two-way interactions.

Variable	Ceftiofur		Danofloxacin	
	MIC Ratio (95% CI)	*p*	MIC Ratio (95% CI)	*p*
Intercept	0.90 (0.30, 2.65)	0.84	0.17 (0.06, 0.53)	<0.01
Unexposed	Referent		Referent	
Exposed	2.58 (0.50, 13.32)	0.26	5.18 (0.97, 27.66)	0.05
Pre-Exposure	Referent		Referent	
1–3 DPE	2.42 (0.99, 5.90)	0.05		
4–7 DPE	4.77 (0.45, 50.81)	0.20		
7–14 DPE	47.41 (11.03, 203.82)	<0.01		
1–5 DPE			64.70 (9.74, 429.85)	<0.01
6–14 DPE			16.81 (0.70, 403.82)	0.08
Exposed × Pre-Exposure	Referent		Referent	
Exposed × 1–3 DPE	0.25 (0.03, 1.82)	0.17		
Exposed × 4–7 DPE	0.87 (0.03, 29.49)	0.94		
Exposed × 7–14 DPE	0.01 (0.001, 0.19)	<0.01		
Exposed × 1–5 DPE			0.03 (0.003, 0.22)	<0.01
Exposed × 6–14 DPE			0.09 (0.003, 2.48)	0.15

**Table 6 antibiotics-12-01019-t006:** Results from significant models for *E. coli* (EC) from individually housed calves where enrofloxacin was the antimicrobial drug of exposure. Coefficients represent minimum inhibitory concentration (MIC) ratios for gentamicin and florfenicol for fixed effects of exposure, days post exposure (DPE) and their two-way interactions.

Variable	Gentamicin		Florfenicol	
	MIC Ratio (95% CI)	*p*	MIC Ratio (95% CI)	*p*
Intercept	0.56 (0.23, 1.37)	0.20	5.28 (1.89, 14.75)	<0.01
Unexposed	Referent		Referent	
Exposed	39.11 (10.17, 150.38)	<0.01	4.36 (1.05, 18.03)	0.04
Pre-Exposure	Referent		Referent	
1–3 DPE	87.79 (10.04, 767.83)	<0.01		
4–14 DPE	77.94 (12.91, 470.42)	<0.01		
1–5 DPE			8.43 (1.14, 62.39)	0.04
6–14 DPE			4.07 (1.12, 14.80)	0.03
Exposed × Pre-Exposure	Referent		Referent	
Exposed × 1–5 DPE			0.06 (0.004, 0.73)	0.03
Exposed × 6–14 DPE			0.54 (0.06, 4.89)	0.58
Exposed × 1–3 DPE	0.01 (0.001, 0.17)	<0.01		
Exposed × 4–14 DPE	0.17 (0.007, 4.11)	0.28		

**Table 7 antibiotics-12-01019-t007:** Predicted minimum inhibitory concentration (MIC) values from significant models for exposed, unexposed and the estimated difference between them (exposed–unexposed) for *E. coli* (EC) from group-housed calves where enrofloxacin was the antimicrobial drug (AMD) of exposure. Minimum inhibitory concentration (MIC) predictions were calculated using the exponentiated coefficients from the model results.

Antimicrobial Drug Tested	Variable Level	Exposed		Unexposed		Exposed–Unexposed	
		MIC (95% CI)	*p*	MIC (95% CI)	*p*	MIC Difference (95% CI)	*p*
Ceftiofur	Pre-Exposure	0.66 (−0.82, 2.14)	0.38	1.29 (−0.37, 2.96)	0.13	−0.64 (−2.85, 1.58)	0.57
	1–3 DPE	0.66 (−0.29, 1.61)	0.18	0.37 (0.26, 0.48)	<0.01	0.29 (−0.67, 1.25)	0.55
	4–14 DPE	11.05 (1.59, 20.49)	0.02	0.43 (0.14, 0.72)	<0.01	10.62 (1.17, 20.07)	0.03

**Table 8 antibiotics-12-01019-t008:** Predicted mean minimum inhibitory concentration (MIC) values from significant models for exposed, unexposed and the estimated difference between them (exposed–unexposed) for *E. coli* (EC) from individually housed calves where enrofloxacin was the antimicrobial drug (AMD) of exposure. Minimum inhibitory concentration (MIC) predictions were calculated using the exponentiated coefficients from the model results.

Antimicrobial Drug Tested	Variable Level	Exposed		Unexposed		Exposed–Unexposed	
		MIC (95% CI)	*p*	MIC (95% CI)	*p*	MIC Difference (95% CI)	*p*
Ceftiofur	Pre-Exposure	2.31 (−0.56, 5.23)	0.12	0.90 (−0.08, 1.86)	0.07	1.42 (−1.63, 4.47)	0.36
	1–3 DPE	1.39 (−0.23, 3.01)	0.09	2.17 (0.59, 3.76)	0.01	−0.78(−3.01, 1.45)	0.49
	4–7 DPE	9.65 (−5.06, 24.36)	0.19	4.28 (−6.89, 15.45)	0.45	5.37 (−12.87, 23.61)	0.56
	8–14 DPE	1.27 (−0.75, 3.29)	0.22	42.52 (−33.86, 118.91)	0.28	−41.26 (−117.59, 35.08)	0.29
Danofloxacin	Pre-Exposure	0.89 (−0.21, 1.98)	0.11	0.17 (−0.02, 0.36)	0.08	0.72 (−0.39, 1.83)	0.21
	1–5 DPE	1.48 (−0.16, 3.13)	0.07	11.06 (−9.70, 31.82)	0.29	−9.58 (−30.40, 11.25)	0.37
	6–14 DPE	0.39 (−0.79, 1.57)	0.52	2.87 (−3.47, 9.21)	0.37	−1.60 (−8.02, 4.82)	0.66
Gentamicin	Pre-Exposure	21.83 (−0.10, 43.76)	0.05	0.56 (0.06, 1.06)	0.03	21.27 (−0.67, 43.21)	0.06
	1–3 DPE	21.83 (−15.09, 58.74)	0.25	49.00 (−62.54, 160.54)	0.39	−27.17 (−144.66, 90.32)	0.65
	4–14 DPE	289.50 (−260.86, 839.86)	0.30	43.50 (−34.66, 121.66)	0.28	246.00 (−309.88, 801.88)	0.39
Florfenicol	Pre-Exposure	22.99 (0.45, 45.52)	0.05	5.28 (−0.15, 10.70)	0.06	17.71 (−5.47, 40.88)	0.13
	1–5 DPE	10.97 (−0.02, 21.96)	0.05	44.48 (−9.11, 98.07)	0.10	−33.51 (−88.22, 21.19)	0.23
	6–14 DPE	50.25 (−9.98, 110.47)	0.10	21.50 (0.53, 42.46)	0.04	28.75 (−35.02, 92.52)	0.38

**Table 9 antibiotics-12-01019-t009:** Percent of enterococci/streptococci (ES) isolates resistant to antimicrobials among exposure statuses, 1–14 days post exposure (DPE), for calves where enrofloxacin was the antimicrobial drug (AMD) used to treat the treated calf. Minimum inhibitory concentration (MIC) breakpoints used to determine resistance, percent resistant, 95% confidence intervals and number of isolates tested (N) are shown.

Antimicrobial Drug Tested	MIC (µg/mL) Breakpoint	Status	N	Group	N	Individual
				% Resistant (95% CI)		% Resistant (95% CI)
Ampicillin	≥16	Unexposed	27	7.4 (0, 17.3)	24	16.7 (1.4, 31.9)
		Exposed	22	13.6 (0, 28.3)	26	3.8 (0, 11.4)
		Treated	17	29.4 (7.1, 51.7)	24	20.8 (4.2, 37.4)
Florfenicol	≥8	Unexposed	27	33.3 (15.6, 51.1)	24	50 (29.6, 70.4)
		Exposed	22	45.5 (25.7, 65.2)	26	42.3 (22.9, 61.7)
		Treated	17	41.2 (17.8, 64.6)	24	37.5 (17.7, 57.3)
Gamithromycin	≥8	Unexposed	27	92.6 (82.7, 100)	24	91.7 (80.4, 100)
		Exposed	22	86.4 (71.7, 100)	26	88.5 (75.9, 100)
		Treated	17	94.1 (82.6, 100)	24	66.7 (47.4, 85.9)
Penicillin	>8	Unexposed	27	11.1 (0, 22.9)	24	16.7 (1.4, 31.9)
		Exposed	22	22.7 (5.2, 40.2)	26	11.5 (0, 24.1)
		Treated	17	23.5 (3.3, 43.7)	24	20.8 (4.2, 37.4)
Tetracycline	≥8	Unexposed	27	100 (-)	24	100 (-)
		Exposed	22	90.9 (78.9, 100)	26	100 (-)
		Treated	17	100 (-)	24	100 (-)
Tiamulin	≥32	Unexposed	27	59.2 (40.4, 78.1)	24	79.2 (62.6, 95.8)
		Exposed	22	81.8 (65.3, 98.3)	26	100 (-)
		Treated	17	70.6 (48.3, 92.9)	24	62.5 (42.7, 82.3)
Tildipirosin	>16	Unexposed	27	0 (-)	24	0 (-)
		Exposed	22	0 (-)	26	0 (-)
		Treated	17	0 (-)	24	0 (-)
Tilmicosin	>16	Unexposed	27	96.3 (89, 100)	24	87.5 (73.9, 100)
		Exposed	22	86.3 (71.7, 100)	26	96.1 (88.6, 100)
		Treated	17	82.4 (63.7, 100)	24	91.7 (80.4, 100)
Tulathromycin	≥32	Unexposed	27	92.6 (82.7, 100)	24	88.5 (76.2, 100)
		Exposed	22	86.3 (72, 100)	26	85.7 (72.8, 98.7)
		Treated	17	76.5 (56.3, 96.6)	24	68 (49.7, 86.3)
Tylosin	≥16	Unexposed	27	92.6 (82.7, 100)	24	95.8 (82.1, 100)
		Exposed	22	86.4 (71.7, 100)	26	88.5 (75.9, 100)
		Treated	17	76.5 (55.7, 97.3)	24	84 (68.1, 98.6)

(-) 95% Confidence Interval could not be calculated.

## Data Availability

All code used for statistical analysis is available at https://github.com/dairyepilab/Article_Breen2023, accessed on 12 May 2023.

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
