# Peer review of "Effect of Group Housing of Preweaned Dairy Calves: Health and Fecal Commensal Antimicrobial Resistance Outcomes"

_antibiotics, 2023, doi:10.3390/antibiotics12061019_

Round 1

Reviewer 1 Report

This study narrates the effect of housing type on the epidemiology of the BRD, development and spread of AMR. The authors indicate the BRD and diarrhea as the important health problem in calves and how frequent occurrence of these problems may lead to overuse of the AMD which can culminate into development of AMR. This manuscript is well written, the introduction section has introduced the topic well, the material and method is well detailed, the result section provide a well narrated information of the results obtained, the discussion concentrate on discussing the results obtained and conclusion was drawn from the results obtained. Few comments are presented in the table below for the authors attention.

Area

Comment

Line 184-185. “….2 EC isolates from UNEXP calves, 3 EC from EXP, and 2 EC from TRT calves could not be regrown……” Same apply to Line 193-194

Can you pleases account on this?

Line 182. “The mean age at sampling was 16.8 days……” and Line 185. “The mean age at sampling was 21.5 day………”

Not clear what is the different between the two mean sampling age.

Line 722. “……spp. Colonies…..”

I think it should be “…….spp. colonies……..”

Table 4 and Table 6

In table 5, the MIC ratio of Ceftiofur 1-3 DPE is statistically significant while 4-14 DPE is insignificant and furthermore, exposure x 1-3 DPE is statistically insignificant while exposure x 4-14 is significant. Same apply to table 6 at exposure x 1-3 DPE and 4-13 DPE. What does this tell us?

Minor edits

Author Response

  1. Line 184-185. “….2 EC isolates from UNEXP calves, 3 EC from EXP, and 2 EC from TRT calves could not be regrown……” Same apply to Line 193-194. Can you please account on this?
    • These lines should have read “.. fecal sample cultures from 3 EXP, and 2 TRT calves did not yield any bacterial colonies after 3 attempts and were excluded”. Some fecal cultures did not yield any isolates identifiable as coli or ES, even after feces from the collected sample were replated 3 times.
  2. Line 182. “The mean age at sampling was 16.8 days……” and Line 185. “The mean age at sampling was 21.5 day”. Not clear what is the difference between the two mean sampling age.
    • The difference is due to the fact that the 16.8 days refers to the mean age of sampling in the Group Housed calves, whereas the 21.5 days refers to the mean age of sampling in the Individual Housed calves. The sampling of EXP and UNEXP calves in Group Housed was independent of that in Individual Housed, so this is where the difference in the mean age of sampling comes from. We mention this as a result because we also made comparisons between resistance proportions between Group Housed and Individual Housed calves, so it is important to know how comparable the groups are.
  3. Line 722. “……spp. Colonies…..”. I think it should be “…….spp. colonies……..”
    • Changed to ‘spp. colonies’
  4. Table 4 and Table 6. In table 5, the MIC ratio of Ceftiofur 1-3 DPE is statistically significant while 4-14 DPE is insignificant and furthermore, exposure x 1-3 DPE is statistically insignificant while exposure x 4-14 is significant. Same apply to table 6 at exposure x 1-3 DPE and 4-13 DPE. What does this tell us?
    • Table 4 Ceftiofur: The only significant coefficient in this model table is Exposed x 4-14 DPE (p<0.01). 1-3 DPE is insignificant at p=0.06.
    • Table 5 Ceftiofur: 1-3 DPE is insignificant at p=0.05. 7-14 DPE and Exposed x 7-14 DPE are significant at p<0.01.
    • Table 6 Gentamicin: 4-14 DPE is significant and Exposed x 4-14 DPE is insignificant. In the presence of an interaction, both main effect and interaction term need to be considered together. The results for the Individual Housed models overall show an increasing resistance over time among the unexposed calves relative to the exposed calves. This was an unexpected finding, and we suggested some possible mechanisms for this effect beginning line 545. I hope I have understood your comment clearly.

Thank you for your comments and consideration.

Reviewer 2 Report

The manuscript was comprehensively written and the results were presented well. Just a few issues with formatting, but I guess it will be taken care of by the editorial team.  

Author Response

Thank you for your comments.

Reviewer 3 Report

The manuscript aimed to evaluate differences in antibiotic use, in the presence of diseases, and in the response of microorganisms to antibiotics in different housing systems of dairy calves. Considering that milk consumption is increasing every year and production needs to production needs to meet consumer demand, the manuscript deals with an extremely relevant topic. The research methodology was well designed, in order to meet the objectives, and is in line with other publications in the area. The obtained results are appropriately presented and discussed, and support the established conclusions.

Some considerations:

Was there any monitoring of zootechnical parameters throughout the study, such as weight gain and feed intake in the different groups of animals?

As the study was conducted on a commercial property, it is understood that there were adult animals. The management of these animals can interfere with the sanitary conditions of the property and the resistance profile of the microorganisms present in the environment. Has there been any control or data survey of the antibiotics used in adult animals, to try to establish some connection with the results observed in the animals used in the research?

Overall Recommendation: Accept after minor revision.

Author Response

  1. Was there any monitoring of zootechnical parameters throughout the study, such as weight gain and feed intake in the different groups of animals?
    • Yes, we measured weight gain, conducted behavioral assessments, and recorded standing and lying times during the study period, however this will be published as part of a different research paper investigating the effects of housing type on the behavior and welfare of dairy calves. This is mentioned on line 613.

  1. As the study was conducted on a commercial property, it is understood that there were adult animals. The management of these animals can interfere with the sanitary conditions of the property and the resistance profile of the microorganisms present in the environment. Has there been any control or data survey of the antibiotics used in adult animals, to try to establish some connection with the results observed in the animals used in the research?
    • We did not analyze any data related to the antibiotic usage of the farm. This would have been interesting information. However, the fact that all the study calves came from this dairy farm, and were subject to the same management, negates the need to control for antimicrobial usage on this farm in adult animals, as any effect would be the same for all of the calves. A similarly designed, multi-site study would benefit hugely from this information.